# Engineered disulfide reveals structural dynamics of locked SARS-CoV-2 spike

Kun Qu [1,2☉], Qiuluan Chen[3☉], Katarzyna A. Ciazynska[1], Banghui Liu[4], Xixi Zhang[4], Jingjing Wang[4], Yujie He[3], Jiali Guan[3], Jun He[3,4], Tian Liu[5], Xiaofei Zhang[3,4,5], Andrew P. Carter[1], Xiaoli Xiong[1,3,4]*, John A. G. Briggs[1,6]*

1 Structural Studies Division, Medical Research Council Laboratory of Molecular Biology, Cambridge, United Kingdom, 2 Infectious Diseases Translational Research Programme, Department of Biochemistry, Yong Loo Lin School of Medicine, National University of Singapore, Singapore, 3 Bioland Laboratory (Guangzhou Regenerative Medicine and Health—Guangdong Laboratory), Guangzhou, China, 4 The State Key Laboratory of Respiratory Disease (SKLRD), CAS Key Laboratory of Regenerative Biology, Guangdong Provincial Key Laboratory of Stem Cell and Regenerative Medicine, Center for Cell Lineage and Development, Guangzhou Institutes of Biomedicine and Health, Chinese Academy of Sciences, Guangzhou, China, 5 Center for Proteomics and Metabolomics, Bioland Laboratory, Guangzhou Regenerative Medicine and Health Guangdong Laboratory, Guangzhou, China, 6 Max Planck Institute of Biochemistry, Martinsried, Germany

☉ These authors contributed equally to this work.
* xiong_xiaoli@gibh.ac.cn (XX); briggs@biochem.mpg.de (JAGB)

**Data Availability Statement:** Cryo-EM maps are deposited in the Electron Microscopy Data Bank https://www.ebi.ac.uk/emdb/ under accession numbers EMD-33453, EMD-33454, EMD-33455, EMD-33456, EMD-33457, EMD-33458, EMD-

## Abstract

The spike (S) protein of SARS-CoV-2 has been observed in three distinct pre-fusion conformations: locked, closed and open. Of these, the function of the locked conformation remains poorly understood. Here we engineered a SARS-CoV-2 S protein construct "S-R/x3" to arrest SARS-CoV-2 spikes in the locked conformation by a disulfide bond. Using this construct we determined high-resolution structures confirming that the x3 disulfide bond has the ability to stabilize the otherwise transient locked conformations. Structural analyses reveal that wild-type SARS-CoV-2 spike can adopt two distinct locked-1 and locked-2 conformations. For the D614G spike, based on which all variants of concern were evolved, only the locked-2 conformation was observed. Analysis of the structures suggests that rigidified domain D in the locked conformations interacts with the hinge to domain C and thereby restrains RBD movement. Structural change in domain D correlates with spike conformational change. We propose that the locked-1 and locked-2 conformations of S are present in the acidic high-lipid cellular compartments during virus assembly and egress. In this model, release of the virion into the neutral pH extracellular space would favour transition to the closed or open conformations. The dynamics of this transition can be altered by mutations that modulate domain D structure, as is the case for the D614G mutation, leading to changes in viral fitness. The S-R/x3 construct provides a tool for the further structural and functional characterization of the locked conformations of S, as well as how sequence changes might alter S assembly and regulation of receptor binding domain dynamics.

33459, EMD-33460, EMD-33461, EMD-33462, EMD-33463, EMD-33464. Associated molecular models are deposited in the Protein Data Bank https://www.ebi.ac.uk/pdbe/ under accession numbers 7xtz, 7xu0, 7xu1, 7xu2, 7xu3, 7xu4, 7xu5, 7xu6. The relationship between dataset name and accession number can be found in S1 Table.

**Funding:** This study was supported by funding from the European Research Council (ERC) under the European Union's Horizon 2020 research and innovation programme (ERC-CoG-648432 MEMBRANEFUSION to JAGB), the Medical Research Council as part of United Kingdom Research and Innovation (MC_UP_A025_1011 to APC; MC_UP_1201/16 to JAGB), Wellcome (210711/Z/18/Z to APC), the Max-Planck Society (to JAGB), and the Natural Science Fund of Guangdong Province (2021A1515011289 to XX). XX acknowledges start-up grants from the Chinese Academy of Sciences and Bioland Laboratory (GRMH-GL). The funders had no role in study design, data collection and analysis, decision to publish, or preparation of the manuscript.

**Competing interests:** The authors have declared that no competing interests exist.

## Author summary

Spike (S) proteins on the surface of SARS-CoV-2 initiate viral infection by binding to cell receptors and mediating the fusion of virus and cell membranes. Several conformations of S have been identified that are thought to exist at different steps of the virus life cycle, for example assembly, receptor-binding and entry. The function of a conformation termed "locked" has not been clearly understood, due to its transience. Here, we engineered a disulfide bond in SARS-CoV-2 S to stabilise the locked conformation for structural and biochemical studies. This allowed us to distinguish two distinct locked-1 and locked-2 conformations in S from the initial SARS-CoV-2 strain, only the locked-2 conformation still exists after introduction of the D614G mutation. Based on structural and biochemical characterizations, we propose that the locked conformations of S prevent the premature opening of the receptor binding domain during virus assembly and egress through intra-cellular compartments. Our engineered S provides a useful tool to facilitate structural research, serological research, and design of immunogens.

## Introduction

The spike (S) protein of coronaviruses is responsible for interaction with cellular receptors and fusion with the target cell membrane [1]. It is the main target of the immune system, therefore the focus for vaccine and therapeutics development. Most candidate Severe acute respiratory syndrome coronavirus-2 (SARS-CoV-2) vaccines utilise S protein or its derivatives as the immunogen [2] and a number of antibodies targeting the SARS-CoV-2 S protein are under development for COVID-19 treatment [3].

The S protein of SARS-CoV-2 is a large, 1273 residue, type I fusion protein, that can be processed by host proteases into two parts, S1 and S2. In the prefusion, trimeric form, S1 forms multiple folded domains. From N-terminus to C-terminus, they are the N-terminal domain (NTD, also called domain A, residues 13–307), receptor binding domain (RBD, also called domain B, residues 330–528), Domain C (residues 320–330 and 529–592), and Domain D (residues 308–319 and 593–699) [4–6]. Neutralising antibodies have been identified that target NTD and RBD [7]. S2 contains the fusion peptide (817–832), S2' cleavage site (811–815), and a fusion peptide proximal region (FPPR) C-terminal to the fusion peptide (previously referred to by us as the 833–855 motif [4]) that we and others have previously suggested may take part in regulating RBD opening [4,8,9] (**Fig 1A**).

SARS-CoV-2 spike protein has been captured in four distinct conformations by cryo-electron microscopy (cryo-EM): three prefusion conformations–locked [4,8–10], closed [5,6], open [5,6]; and one post-fusion conformation [9]. The different conformations of S expose different epitopes [11] and therefore induce different immune responses [12]. The different conformations must therefore play distinct roles in the infection cycle.

In the locked and closed conformations, the three copies of RBD in each spike trimer lie down on the top of the spike such that each RBD interacts *in trans* with the NTD of the neighbouring S protomer, hiding the receptor binding site. In the locked conformation, the three RBDs are in very close proximity at the 3-fold axis, and a disulfide-bond stabilized helix-turn-helix motif (FPPR motif) is formed below the RBD that can sterically hinder RBD opening [4,8–10]. Additionally, a linoleic acid is bound within the RBD and this ligand has been proposed to stabilize the locked conformation [10]. The locked conformation is adopted in solution by the Novavax NVX-CoV2373 vaccine candidate [8]. In structures of the locked conformation, both NTD and RBD are well resolved, suggesting that the locked spike has a

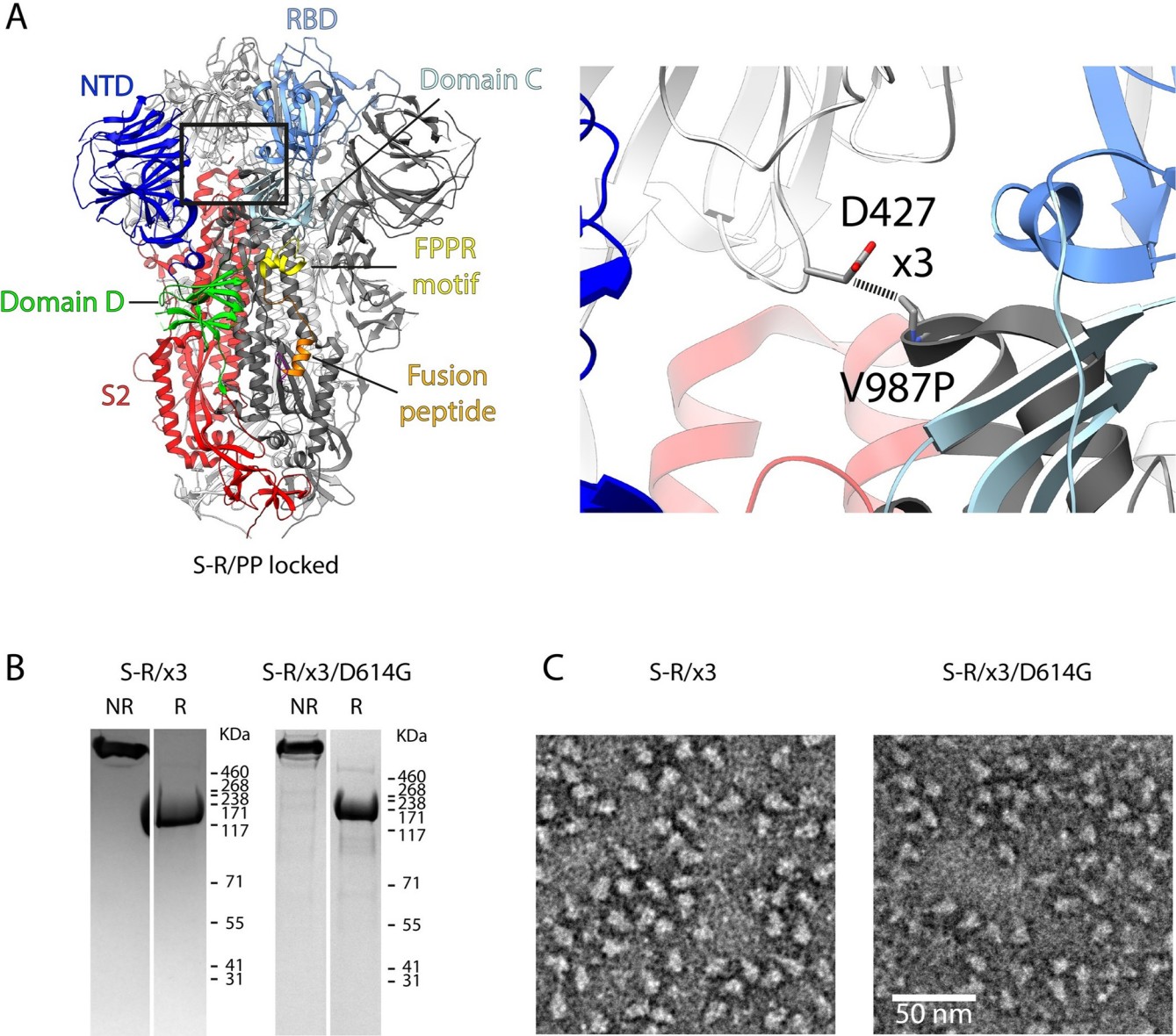

**Fig 1. Design of the x3 disulfide bond to stabilise the SARS-CoV-2 spike in the "locked" state.** (A) Structure of the S-R/PP spike in the locked state (PDB: 6ZP2). Structural domains are coloured and labelled. The box indicates the location of the zoomed-in view in the right panel. Residues 427 and 987 that were mutated to cysteine to form the x3 disulfide bond are indicated. (B) Coomassie-stained SDS-PAGE gels to assess expression of the S-R/x3 and S-R/x3/D614G S proteins and confirm formation of a disulphide bond. NR or R indicates protein samples were prepared in non-reducing or reducing conditions. (C) Negative stain EM images of the purified S-R/x3 and S-R/x3/D614G spikes.

rigid overall structure. In the closed conformation, the RBDs are slightly further apart than in the locked conformation and the FPPR motif is not folded [4,8–10]. As a result, the NTD and RBD exhibit considerable dynamics and are less well resolved than the core part of the protein in cryo-EM structures. Closed SARS-CoV-2 spike usually coexist with spikes exhibiting an open conformation [5,6]. In the open conformation, one or more RBDs are raised up to expose the receptor binding loops. ACE2 or antibody binding to these loops appears to maintain the spike in an open conformation [13,14]. It is believed that opening of the cleaved spike ultimately leads to structural transition into the postfusion conformation. The virus uses the

dramatic structural refolding from the prefusion conformation to the lowest-energy postfusion conformation to drive membrane fusion [15].

Among the observed conformational states of SARS-CoV-2 spike, the locked conformation is unique in that it has not been observed for the highly related SARS-CoV-1 spike. Other coronavirus spikes do have locked-like conformations featuring FPPR motifs that preventing RBD opening, but for many of them, stochastic formation of an open conformation has not been observed [16]. The early cryo-EM studies of purified SARS-CoV-2 spike ectodomain did not identify the locked conformation, and we previously found that only a small fraction of trimeric ectodomains adopt the locked conformation [4], suggesting this conformation is transient. However, the locked conformation continued to be identified in other cryo-EM studies: spikes in locked conformation have been identified in purified full-length spike proteins including the transmembrane region [8,9]; insect-cell-expressed spike ectodomains can also adopt the locked conformation, perhaps due to stabilizing lipid in the insect cell media [10]. However, curiously, the locked conformation was not observed in the high-resolution structures of spike proteins on virions, where only closed and open conformation spikes were observed [17].

Considering the above observations, the role of the locked conformation in the SARS-CoV-2 life cycle is unclear. Here, based on our previous locked spike structure [4], we engineered a disulfide bond which stabilizes the SARS-CoV-2 spike in the locked conformation. This construct provides a tool to characterize the structure and dynamics of the locked form of S and to assess how mutations and other environmental factors could affect its structure and function. Our data reveal multiple distinct locked conformations and suggest that structural dynamics of domain D modulates the conformational state of the SARS-CoV-2 spike. Newly obtained information regarding the "locked" spike allowed us to consider its functional role in the context of SARS-CoV-2 life cycle.

## Results and discussion

### Design of a spike protein ectodomain stabilized in the locked conformation

We did not previously observe a locked conformation for S-R (S with a deletion at the furin cleavage site leaving only a single arginine residue), while for S-R/PP (S-R additionally containing the widely-used, stabilizing double-proline mutation) the locked conformation was rare, <10% of S trimers when imaged by cryo-EM [4]. The scarcity of the locked conformation for the expressed ectodomain makes it difficult to study. To overcome this challenge, based on the locked SARS-CoV-2 S-R/PP structure [4], we engineered cysteines residues replacing positions D427 and V987 to generate a new disulfide-stabilized S protein S-R/x3. We predicted that S-R/x3 should be predominantly in the locked conformation (**Fig 1A**). Under non-reducing conditions, S-R/x3 ran exclusively as trimer in SDS-PAGE gel, and the trimer is converted to monomer under reducing conditions (**Fig 1B**). This behaviour is similar to our previously engineered S-R/x2, suggesting efficient formation of disulfide bond between the 2 engineered cysteines during protein expression. Negative stain electron microscopy (EM) images of purified S-R/x3 show well-formed S trimers (**Fig 1C**). We additionally introduced a D614G substitution into the S-R/x3 spike to obtain S-R/x3/D614G. Similar to S-R/x3, purified S-R/x3/D614G ran as disulfide-linked trimers in SDS-PAGE gel under non-reducing conditions, and negative stain EM images show well-formed trimers (**Fig 1B and 1C**).

### Biochemical properties of the x3 spike protein

We tested the sensitivity of the x3 disulfide bond to reduction by dithiothreitol (DTT), and compared this to the sensitivity of the previous described x2 disulfide bond, which stabilizes

the spike in the closed state [4]. After 5 min of incubation with 2.5 mM DTT, the x3 disulfide in both S-R/x3 and S-R/x3/D614G spikes was substantially reduced, and 5 min of incubation with 20 mM DTT was sufficient for almost complete reduction (**Fig 2A**). In contrast, a 60 min incubation with 20 mM DTT was needed to achieve near-complete reduction of the x2 disulfide bond (**Figs 2A and S1A**). The x3 and x2 disulfides both include C987 in S2, therefore, different local chemical environments of the other engineered cysteines at position 427 and 413 respectively, are likely to be responsible for the difference in susceptibility to reduction by DTT.

We tested the receptor binding properties of S-R/x3 and S-R/x3/D614G in the absence and presence of DTT. In the absence of DTT, S-R/x3 and S-R/x3/D614G bind ACE2-Fc with weak responses (<0.4) and low affinities (>100 nM) (**Fig 2B**), while under similar conditions, S-R and S-R/PP bind ACE2-Fc with maximum responses between 0.9–1.0 and affinities of ~1 nM [12]. In the presence of 20 mM DTT, S-R/x3 and S-R/x3/D614G bound ACE2-Fc with maximum responses of 0.8–1.0 and affinities in the low nano-molar range (**Fig 2B**). In contrast, likely due to the inefficiency of x2 disulfide bond reduction, S-R/x2 and S-R/x2/D614G only moderately increased responses and affinities under reducing conditions (**Fig 2B**). These observations are consistent with our prediction that the x2 and x3 disulphide bonds prevent RBD opening and thereby prevent strong receptor binding, and that reduction of the x3 disulphide by DTT permits RBD opening and receptor binding.

Entirely in line with the receptor binding data, we found that the antibody CR3022, which binds a cryptic epitope exposed only in the open RBD, is only able to bind S-R/x3 and S-R/x3/D614G spikes in the presence of DTT, and does not bind S-R/x2 and S-R/x2/D614G spikes (**S1B Fig**).

## Cryo-EM Structures of S-R/x3

We performed cryo-EM of S-R/x3 trimers as described previously for S-R/x2 [4]. Consistent with the predicted effect of the engineered disulfide bond, 3D classification of S-R/x3 particles showed that the majority of the particles (76%) are in the locked conformation, while 23% of the particles are in the closed conformation (**S2 Fig**). Consensus refinement imposing C3 symmetry of all the locked spike particles obtained a 2.6 Å resolution map (**S2 Fig**). The x3 disulfide bond appears to be radiation sensitive, and the x3 disulfide density is only visible in the cryo-EM density derived from early exposure frames (**S3 Fig**). Further 3D classification after symmetry expansion revealed that locked S-R/x3 spike protomers adopt two distinct conformations that differ primarily at domain D (**Figs S2 and 3A and 3B**), we will refer to these conformations as "locked-1" and "locked-2". Classification revealed that locked S-R/x3 S trimers were formed by all four possible combinations of locked-1 and locked-2 protomers (the combinations 111, 112, 122 and 222 (**S2 Fig**)).

In the locked-1 conformation, residues 617–641 in domain D form a large loop, of which residues 632–641 can be clearly identified in the cryo-EM density while residues 619–631 are disordered and not resolved (**Fig 3A**). In the locked-2 conformation, residues 617–641 in the locked-2 protomer are ordered and can be clearly identifiable in cryo-EM density (**Fig 3B**). Both locked-1 and locked-2 conformations are characterised by: structural rigidity (showing minimal local dynamics in NTD and RBD (**S4 Fig**); RBDs are tightly clustered around the three-fold symmetry axis (**S5A and S5B Fig**); linoleic acid is bound into the described pocket in the RBD (**Fig 3A and 3B**); and residues 833–855 are folded into a helix-loop-helix fusion peptide proximal region (FPPR) motif (**Fig 3A and 3B**). These resolved features are consistent with our previously observations of the locked structure [4], further confirming that insertion of the x3 disulfide bond successfully stabilizes the locked conformation during protein

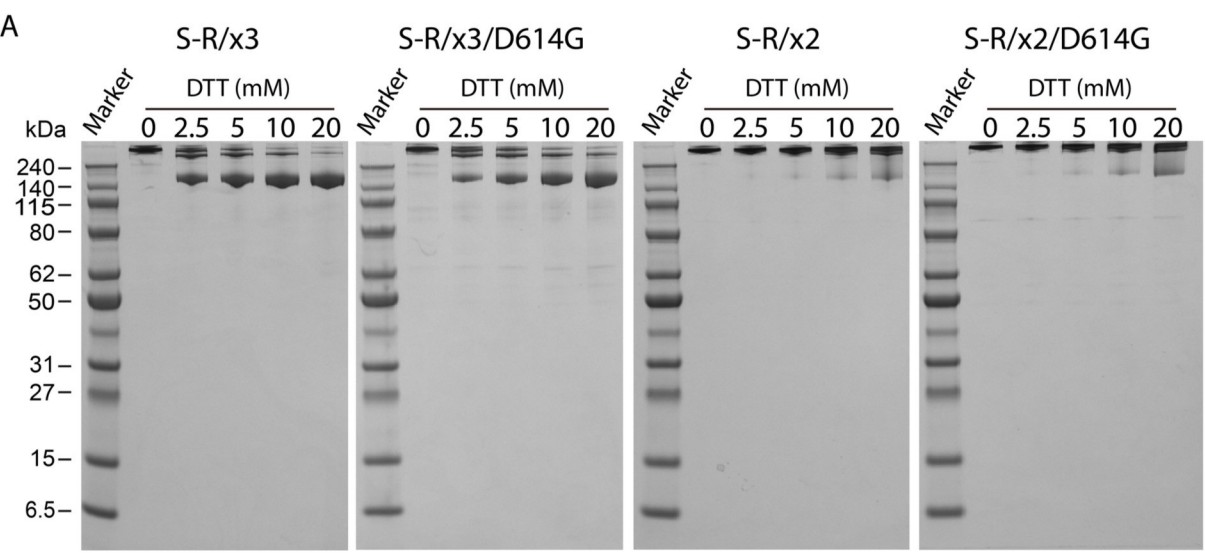

B Binding to ACE2-Fc

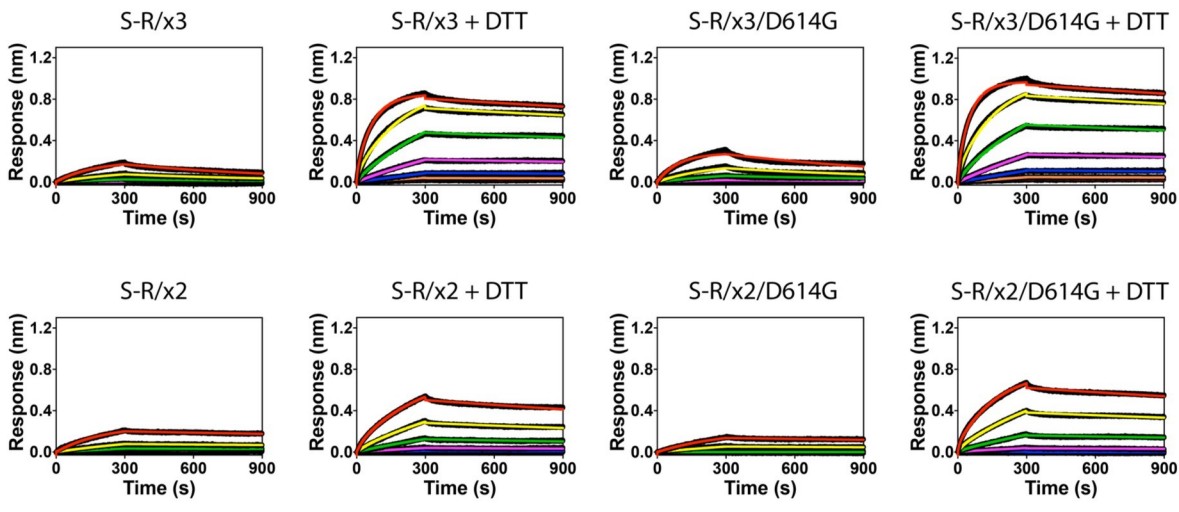

| Ligand | Analyte | $k_{on}$ ($M^{-1}$ $s^{-1}$) | $k_{off}$ ($s^{-1}$) | $K_{Dkin}$ (nM) |
|---|---|---|---|---|
| ACE2-Fc | S-R/x3 | $3.23\times10^3$ | $12.48\times10^{-4}$ | 385.9 |
| | S-R/x3+DTT | $113.40\times10^3$ (fast) | $1.68\times10^{-4}$ | 1.48 |
| | | $9.47\times10^3$ (slow) | | 17.73 |
| | S-R/x3/D614G | $6.52\times10^3$ | $9.86\times10^{-4}$ | 151.23 |
| | S-R/x3/D614G+DTT | $136.60\times10^3$ (fast) | $1.61\times10^{-4}$ | 1.18 |
| | | $11.84\times10^3$ (slow) | | 13.61 |
| | S-R/x2 | $3.06\times10^3$ | $2.21\times10^{-4}$ | 72.32 |
| | S-R/x2+DTT | $18.63\times10^3$ (fast) | $3.22\times10^{-4}$ | 17.28 |
| | | $2.20\times10^3$ (slow) | | 146.56 |
| | S-R/x2/D614G | $2.06\times10^3$ | $2.33\times10^{-4}$ | 112.84 |
| | S-R/x2/D614G+DTT | $75.61\times10^3$ (fast) | $2.34\times10^{-4}$ | 3.09 |
| | | $3.70\times10^3$ (slow) | | 63.23 |

**Fig 2. Biochemical properties of purified x3 spikes comparing to x2 spikes.** (A) Reduction of x3 and x2 disulfide bonds under native conditions by 5 min incubation with indicated concentrations of DTT, reactions were stopped by excess amount of iodoacetamide before SDS-PAGE. (B) binding of ACE2-Fc to different spike proteins in the absence and presence of 20 mM DTT. Spike proteins were serial diluted to 1500, 500, 166.7, 55.6, 18.5, 6.17, 2.06 and 0 nM and used as analytes in the assays. Kinetic parameters ($k_{on}$, $k_{off}$) and dissociation constants derived from kinetic analyses ($K_{Dkin}$) are summarised in the table.

expression and purification. We observed density consistent with the presence of the pigment biliverdin in the reported NTD binding pocket [18] in all our spike structures (**Fig 3**), consistent with the greenish color of the purified protein. We confirmed the presence of biliverdin by mass spectrometry (**S6 Fig**).

## Cryo-EM Structures of S-R/x3/D614G

Introduction of the D614G substitution into SARS-CoV-2 S spike ectodomain constructs with the double proline stabilizing modification has been reported to lead to a higher fraction of purified S in the open conformation [19,20]. Consistent with this, we found that 63% of purified S-R/D614G spikes adopted an open conformation (the remainder are in a closed conformation) (**S2 Fig**), compared to ~18% of S-R spikes in the open conformation [4]. We previously speculated that the D614G substitution alters the structure of the locked conformation, modulating spike structural dynamics [4]. Therefore, we determined the structure of the S-R/x3/D614G spike. The proportion of locked conformations is greatly reduced compared to S-R/x3, constituting only ~19% of total particles (**S2 Fig**). Classification did not identify two different locked conformations of the S-R/x3/D614G –all locked spikes were in the locked-2 conformation. As in the S-R/x3 locked-2 structure, residues 617–641 are ordered, and linoleic acid is bound (**Fig 3D**) confirming that the D614G spike retains the ability to bind linoleic acid, a molecule that has been suggested to be important for regulating conformational state of the spike [10]. Despite the presence of the x3 disulfide bond, 81% of the particles are in a closed conformation. The closed conformation accommodates the disulfide bond by motion of the RBD towards S2 (**S5G Fig**).

## Structural transition of locked spike to closed conformation and the effect of low pH

The S-R/x2 spike can be stored at 4˚C for at least 30 days without significant loss of particles by negative stain EM [4]. We observed that the S-R/x3 spike can also be stored at 4˚C for 40 days retaining well-formed trimers in negative stain EM. We performed cryo-EM imaging of the remaining trimers, finding that they had transitioned into the "closed" conformation (**S2 Fig**), and that linoleic acid was no longer bound in the RBD. This suggests that both the locked-1 and locked-2 conformations are metastable at pH 7.4, and spontaneously transition into the closed conformation. It is consistent with the general observation that most published ectodomain structures (not stabilized by the x3 crosslink) adopt the closed/open conformations rather than the locked conformation. We adjusted the buffer pH of the now "closed" spike to pH 5.0 and incubated overnight before determining its structure by cryo-EM. We found that low pH treatment had converted most of the closed spike to a conformation similar to the locked-2 conformation (**Figs 3F and S2**), however, in the acid reverted locked-2 structure the linoleic acid binding pocket is unoccupied (**Fig 3F**).

## The structure of domain D differs in locked-1 and locked-2 conformations

The locked-1 and locked-2 conformations differ primarily at residues 617–641 within domain D, and this structural perturbation is associated with differences in the FPPR motif and in

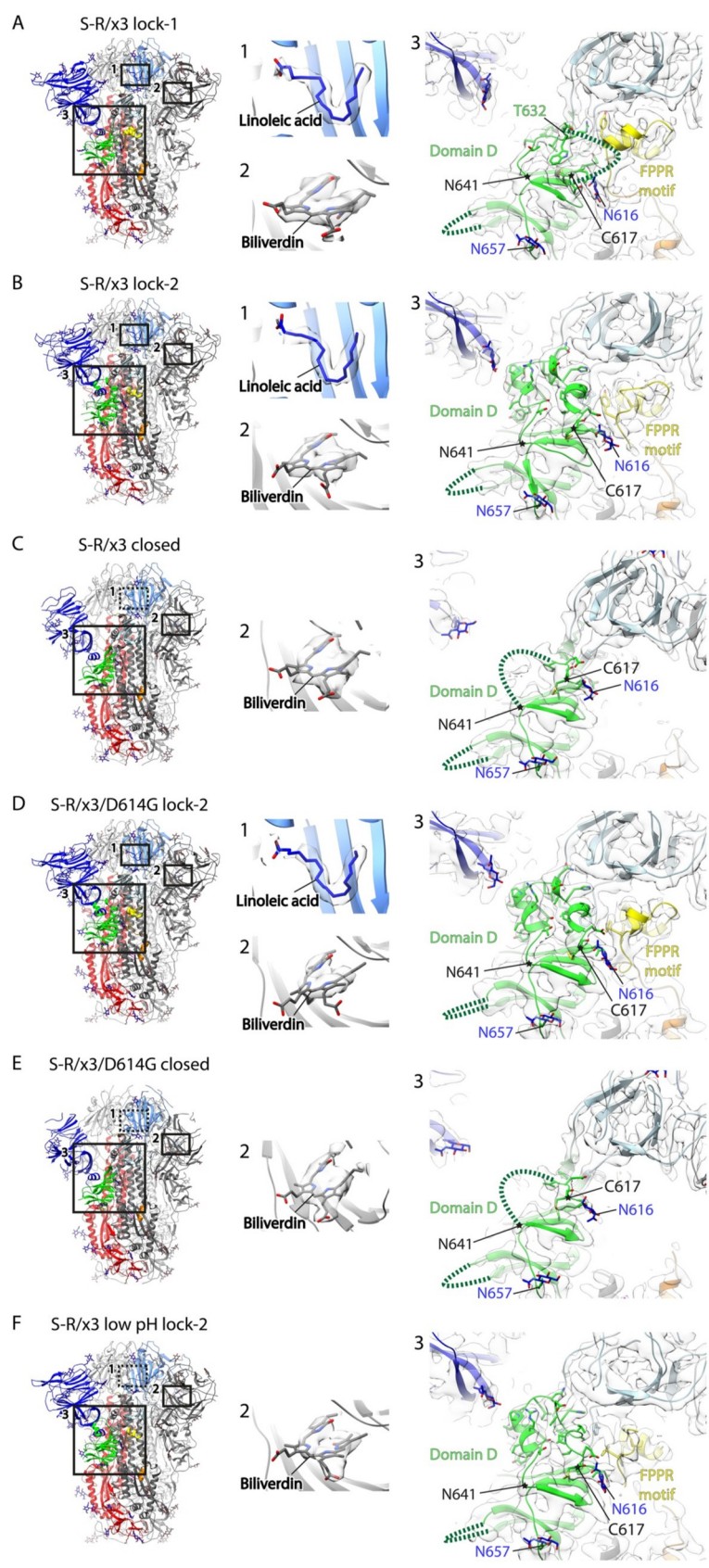

**Fig 3. Structural features of spike proteins in different conformations as determined by cryo-EM.** (A)—(F) the left panels show side views of indicated spike proteins and conformational states, the structural domains are coloured as in Fig 1. The numbered boxes indicate the zoomed-in views as shown in the middle and right panels, dashed boxes indicate absence of bound factors. The middle panels show cryo-EM densities of bound factors; the right panels show cryo-EM densities of structural regions around domain D. Disordered loops are indicated by dashed lines, glycans are shown as stick representations in blue. The region between C617 and N641 (marked with asterisks) changes conformation between the different structures.

residues around 318–319 which are located in the junction region connecting domain C and D (**Figs 3A, 3B, 4A and 4B**). In the S-R/x3 locked-1 structure, the long sidechains of R634 and Y636 interdigitate with the sidechains of F318 and Y837 through cation-π and π-π stacking interactions (dashed red lines in **Fig 4A**. In this interaction, R634 in domain D is sandwiched between F318 of the domain C/D junction region, and Y837 of the FPPR motif, bridging these three structural features. These interactions appear to be further stabilized by hydrophobic contacts between the aromatic sidechains of F318 and Y636 and the domain D hydrophobic core. This arrangement positions loop residue R634 approximately 9 Å above D614 (**Fig 4A**). It appears likely that this zip-locking interaction not only maintains residues 617–641 in a loop structure but also restrains motion of domain C relative to domain D.

In the locked-1 structure, we and others observed a salt bridge between D614 and K854 within the FPPR motif [4,9,10] (**Fig 4A**). We suggest that substitution of negatively charged D614 to neutral G alters the local electrostatic interactions, preventing R634 binding between F318 and Y837, and triggering the 617–641 loop refolding. This would provide an explanation for the absence of the locked-1 conformation in S-R/x3/D614G.

In the locked-2 structure, 617–641 loop refolds into two short alpha helices, and the side-chains of F318 and R319 at the domain C/D junction reorient (**Fig 4B**). These structural changes allow formation of new electrostatic cation-π interactions between R319 and aromatic residues W633 and F592 (dashed red lines in **Fig 4B**) and formation of hydrophobic interactions between residues in the 617–641 loop and the hydrophobic core of domain D formed by the beta-sheet structure (**Fig 4B**). In the acid-reverted locked-2 structure of S-R/x3, F592 is no longer within cation-π interaction distance of R319 and instead forms cation-π interactions with H625 which is positively charged under low pH (**Fig 4D**).

## Comparison to other spike structures

Most SARS-CoV-2 spike structures deposited in the PDB are in the open or closed conformations and have a disordered FPPR motif and unresolved 617–641 region (https://www.ebi.ac.uk/pdbe/pdbe-kb/protein/P0DTC2). Only a few cryo-EM studies have captured S trimers in locked conformations, including in purified full-length spike protein with or without the PP mutation [8,9] (**S8A, S8B and S8H Fig**); soluble S-GSAS/PP trimers (where the furin cleavage site is replaced with a GSAS sequence) purified from insect cells [10] (**S8C and S8D Fig**); soluble S-GSAS/PP trimers purified from mammalian cells [14,21] (**S8E–S8G Fig**), and our previous study of soluble S-R/PP and S-R/PP/x1 [4]. In all of these cases the construct used had a D at position 614 and the locked-1 conformation was observed (**S8A–S8H Fig**). Among these studies, the percentage of particles adopting the locked conformation and the dynamics of the purified spike vary. The reason why locked conformations are observed under some conditions but not others is not completely clear. Detergent in the purification procedure for the full-length spike [8,9] and high detergent concentration during the EM grid preparation [21] could prevent certain species of spike from being imaged. Other factors such as the presence of membrane anchor in the full-length spike [8,9]; differences in expression conditions—added lipid in insect cell media, low pH condition of insect cell media [10]; and differences in

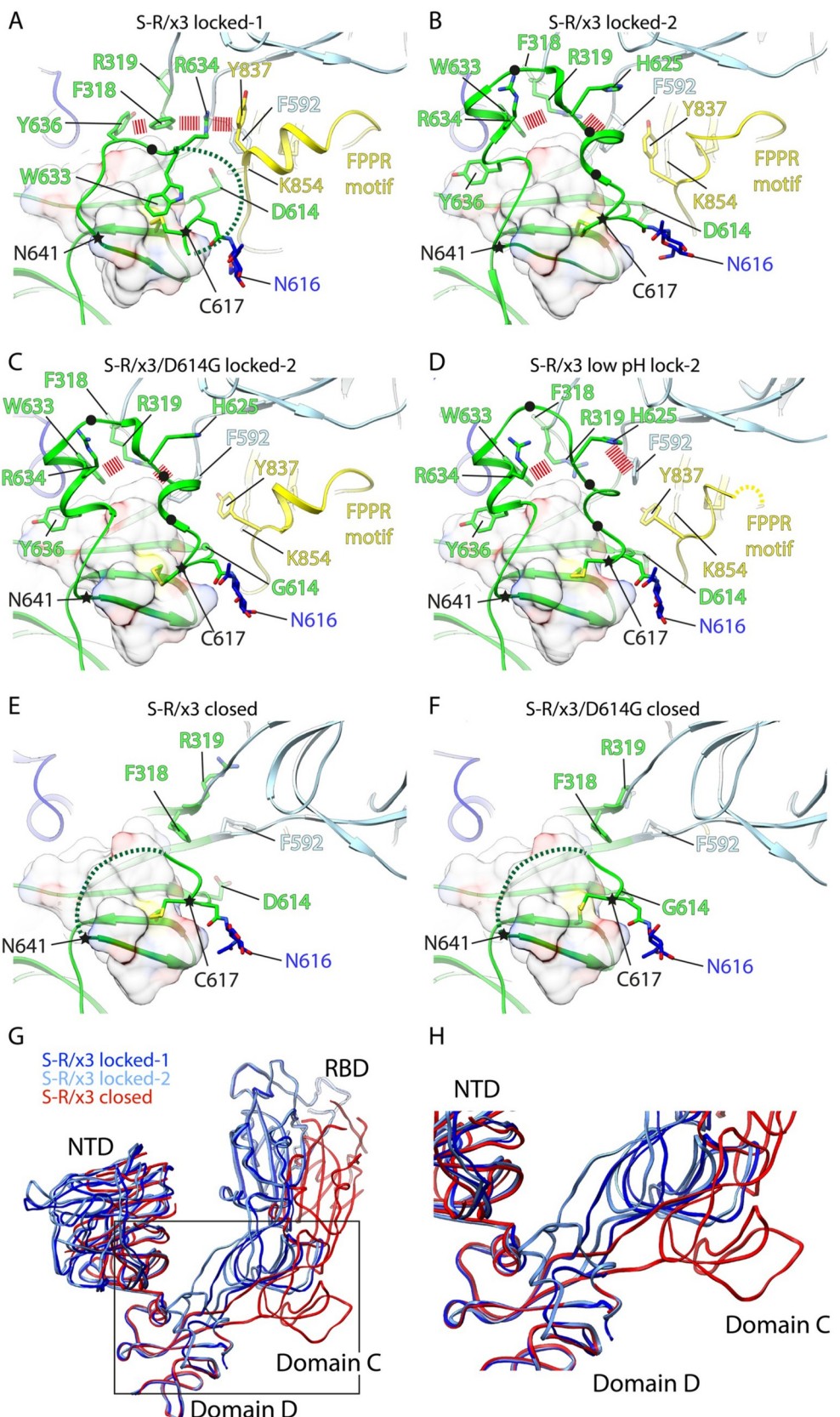

**Fig 4. Structural changes in domain D between spikes in locked-1, locked-2 and closed conformations.** (A)—(F) structures of domain D in S-R/x3 and S-R/x3/D614G spikes of different conformations. Domain C is in light blue, Domain D green, FPPR yellow. Selected amino acid sidechains are shown and key amino acid sidechains are marked. The disordered loops are represented by dashed lines. C617 and N641 are marked with stars to highlight the dynamic region in between. π-π and cation-π interactions stabilising locked conformations are highlighted with dashed lines. Positions of hydrophobic residues with omitted side chains interacting with the domain D hydrophobic core are marked by black dots. The hydrophobic core formed by the domain D beta sheet is shown as a transparent molecular surface. (G) overlay of the S1 backbone from different conformations of the S-R/x3 spike. (H) the region within the box in panel (G) is zoomed to highlight the movements of the domain C/D junction and domain C between locked and closed conformations.

reagents as concluded by Xu et al [14], may have all played a part. However, we speculate that the primary reason that different studies captured this protein in varied conformations is the inherent instability and dynamic nature of certain spike conformations combined with differences in sample preparation methods during purification and imaging. This conclusion justifies the need for development of methods to stabilize distinct conformations of SARS-CoV-2 spike.

The structure of the full-length 614G spike protein (PDB:7krq) [22] is in an intermediate conformation between locked-2 and closed conformations (**S7B** and **S8I and S8J** Figs). The FPPR motif in this structure is ordered, but linoleic acid is missing in the RBD and the 617–641 loop only partially contacts the domain D hydrophobic core. We proposed that the capture of this intermediate form reflects rapid transition of the D614G spike away from the locked-2 conformation, consistent with our observations that the majority of S-R/D614G spike is open (**S2 Fig**) and that even in the presence of x3, only a minority of S-R/x3/D614G spikes adopt the locked-2 conformation.

Zhou and colleagues investigated the effect of low pH on ACE2 and antibody binding to SARS-CoV-2 spike protein [23]. The best-resolved protomer in their all RBD "down" structure at pH 5.5 (6xm5) is similar to our acid-reverted S-R/x3 structure: the 617–641 loop is ordered to contact the domain D hydrophobic core and a cation- π interaction is formed between R319 and W633. Unlike in our structure, the FPPR motif is still showing some dynamics and is not retracted back to be below domain C (**S8L Fig**), this affects the positioning of F592 so that it is not interacting with H625. Therefore, 6xm5 is likely to have captured an intermediate state in the low pH promoted structural transition to locked-2 conformation. In line with our conclusion, this observation demonstrates that low pH may not be sufficient to revert the 614D spike in the closed conformation back to the locked-1 conformation.

## The functional role of the locked-1 and locked-2 structures

Despite their differences, locked-1 and locked-2 conformations have in common that the 617–641 loop interacts with the domain D hydrophobic core and with residues at the domain C/D junction such as F318, R319 and F592 (**Fig 4A–4D**). In contrast, in the closed and open spike structures (independent of whether 614 is D or G) these interactions are lost, and the 617–641 loop density is disordered (**Figs 3C, 3E, 4E and 4F**). Superposition of locked and closed structures indicates that the loss of interactions at the domain C/D junction leaves the junction unrestrained, allowing domain C and the RBD to move relative to domain D and the NTD (**Figs 4G and 4H and S7B**) and allowing the spike to open. Transitions from the locked towards the closed and open positions of the RBD therefore correlates with the structural changes in domain D.

Low pH is able to revert the closed spike towards the locked-2 conformation by restoring interactions in domain D and the domain C/D junction (**Fig 4D**) but cannot revert the S-R/x3 closed spike to the locked-1 conformation as the complex interdigitation involving 617–641

loop residues (**Fig 4A**) cannot be reformed. The absence of the locked-1 conformation in S-R/x3/D614G suggests to us that in the absence of the interaction between D614 and R634/K858 the interdigitation interaction was not able to form during the 614G spike synthesis.

Based on the coexistence of S-R/x3 locked-1, locked-2 and closed conformations in freshly purified protein, and the spontaneous transition of S-R/x3 locked conformations to closed conformation in our storage experiment (**S2 Fig**), we speculate that the S-R/x3 spike protein is synthesized in the locked conformations and transitions to closed conformation due to instability of locked conformations at neutral pH. Similarly, the observations that the locked-1 conformation is more prevalent than the locked-2 conformation in S-R/x3 (**S2 Fig**), and that the majority of purified S-R/x3/D614G spikes (which can only adopt the locked-2 conformation) have transitioned away from the locked conformation (**S2 Fig**), suggests that the locked-2 spike with the rigidified domain D is more prone to transition into the closed conformation under neutral pH than the locked-1 spike with interdigitated domain D. Transition of domain D away from the locked conformations removes the structural restraint on motion of domain C and the RBD. Structural transitions between locked, closed and open conformations are complex, and although we suggest that changes in domain D are a dominant factor in regulating transition between locked and closed conformations, other factors, including linoleic acid binding and furin cleavage [24], may also play a role.

The absence of the locked-1 conformation with interdigitated domain D in constructs containing the D614G substitution is likely due to the change of electrostatic properties caused by the D614G substitution. Instead, D614G-containing spikes adopt the locked-2 conformation which is more prone to transition away from the locked state towards closed and open conformations (**Fig 5A**). These phenomena explain why studies of 614G SARS-CoV-2 spikes without stabilizing mutations failed to capture the locked conformations, and it is consistent with observations that the 614G spike tends to adopt an open conformation [19,20,25,26]. It should also be noted that many studies of D614G-containing spikes also included the double proline mutation [19,20,25] which further promotes the open state.

Beta-coronaviruses have recently been shown to assemble in the low pH, high lipid environment of the ERGIC, and to egress through somewhat acidic compartments of the endomembrane system that are adapted by viral accessory proteins [27] (**Fig 5B**). Using x3 stabilized 614D and 614G spikes, we found both nascent locked-1 and locked-2 conformations bind lipid, and that low pH is able to revert the 614D spike towards the locked-2 conformation. Given that both low pH and lipid binding favour locked conformations, we speculate that during virus assembly, both the 614D and 614G spikes are in locked conformations, and that this provides a mechanism to prevent premature transition into open or post-fusion conformations during virus assembly and egress. Once viruses are released into the neutral pH environment outside of the cell, the spike transitions to the closed or open conformations (**Fig 5B**). The 614D spike may transition from locked-1 to locked-2 to closed to open, while the 614G spike can only adopt the locked-2 conformation, and exhibits different structural dynamics and a higher tendency to open. We suggest that the shifted locked to closed equilibrium results in a higher tendency to open (**Fig 5A**). This may work in tandem with stabilization of the open conformation by the D614G mutation [25,28,29] or a reduction in spontaneous shedding of the S1 subunit [30,31]. The resulting change in virus life cycle is not fully understood and could potentially affect receptor binding, cell entry and antigen presentation. Nevertheless, the D614G change provided a transmission advantage leading to global dominance of the mutant virus based on which all current variants of concern evolved [32]. Further study of the structural transitions between locked-1, locked-2 and closed conformations of the spike is required, and can be facilitated by the x3 construct described here.

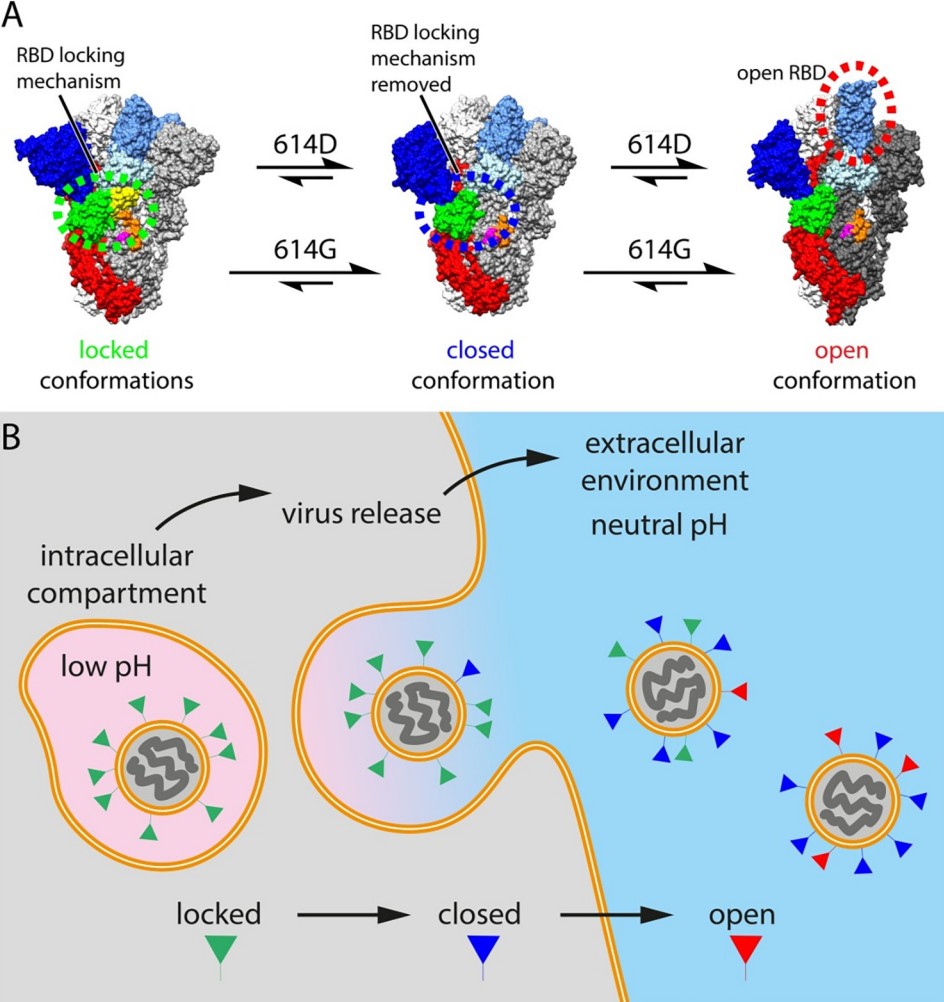

**Fig 5. SARS-CoV-2 virus particle release and the accompanied structural transitions of surface spike protein.** (A) the three prefusion conformational states observed for the SARS-CoV-2 spike protein and factors that influence the structural transitions between the conformational states. Data in this study suggests that the D614G mutation may modulate locked to closed transitions. Data in the literature suggests that the D614G mutation may also modulate closed to open transitions and the stability of the open form (see main text). (B) schematic diagram illustrating the release of nascent SARS-CoV-2 virus particles from the cell. Green, blue or red triangles indicate the predicted conformational states of spike proteins as the virus travel through the acidic (pink) intracellular compartment before it was released into the neutral (blue) extracellular environment.

## Methods

### Expression constructs

Expression constructs were generated essentially as described in Xiong et al. 2020 [4]. Starting from the S-R construct [4], introduction of cysteines to form x3 disulfide and introduction of the D614G substitution were carried out using Q5 polymerase PCR with primers containing desired substitutions, followed by In-Fusion HD (Takara Bio) assembly. The previously reported construct consisting of the ectodomain of human angiotensin-converting enzyme 2 (ACE2, residues 19–615) fused with a Fc-tag [33] was cloned into a pCDNA3.1 vector using In-Fusion assembly.

## Protein production and purification

Spike proteins were expressed in Expi293 cells and purified by metal exchange chromatography exactly as described in Xiong et al. 2020 [4]. Purified spike proteins were quality checked by negative stain EM. To purify ACE2-Fc protein, 500 μg of plasmid was incubated with 1350 μg of polyethylenimine for 10 min, and then transfected into 500 ml of 293F cells at 3 million/ml. The transfected cells were cultured at 33˚C. Cell culture supernatant was collected and clarified at day 6 post transfection and loaded onto a 5 ml Hitrap protein A column (Cytiva). The column was washed with 100 ml PBS and eluted with citric acid buffer (100 mM, pH 3.0). The eluted fractions were collected into 1 M Tris-HCl (pH 8.0) buffer and neutralized. Purified ACE2-Fc was concentrated and buffer exchanged into PBS by a 50-kDa MWCO ultra centrifugal filter (Millipore).

## Negative staining EM

3 μl of proteins (~0.05 mg/ml) diluted with PBS buffer were deposited onto carbon-coated grids (EMS CF200-Cu) glow-discharged for 15 seconds at 25 mA in air. After 60 s incubation, excess proteins were wicked by filter paper. Grids were washed once in buffer, and stained twice in Nano-W stain (Nanoprobes) with blotting in between. The grids were air dried on filter paper and imaged using a Tecnai T12 Spirit operated at 120 kV. Micrographs were taken in Digital Micrograph (Gatan).

## Disulfide bond reduction under native conditions

S-R/x2, S-R/x2/D614G, S-R/x3, and S-R/x3/D614G spike proteins were incubated with 0, 2.5, 5, 10, and 20 mM DTT for 5, 20 or 60 min at room temperature in PBS. Reactions were stopped by addition of 55 mM iodoacetic amide for 10 min in the dark at room temperature. Reaction mixtures were mixed with 4× non-reducing loading buffer and were analysed by SDS-PAGE.

## Biolayer interferometry

Binding affinities of ACE2-Fc or CR3022 IgG towards SARS-CoV-2 spikes were assessed by BLI on an Octet RED96 instrument (FortéBio, USA) following essentially the same protocol as previously described [12]. All steps were performed at 25˚C and at a shaking speed of 1000 rpm. All reagents were formulated in PBS-TB buffer (PBS with 0.02% v/v Tween-20 and 0.1% w/v BSA). Before the experiments, all biosensors were pre-equilibrated in the PBS-TB buffer for 10 min. ACE2-Fc and CR3022 were diluted to 11 μg/mL to be immobilized on Protein A biosensors (FortéBio, USA) to levels of ~1.2 and ~1.6 nm respectively. After a 60 s baseline step in PBS-TB, the ligand-loaded biosensors were submerged in spike protein solutions (3 fold serial diluted from 1500 to 2.06 nM for ACE2-Fc binding or 2 fold serial diluted from 200 to 3.13 nM for CR3022 binding) for 300s and then transferred into PBS-TB for 600 s to measure spike association and dissociation kinetics. To assess the effect of DTT, equilibration, association and dissociation steps were performed in PBS-TB buffer supplemented with 20 mM DTT. The biosensors were regenerated by 10 mM glycine (pH 2.0) between subsequent experiments. Data were aligned, inter step corrected to the association step and further analyzed using the Data Analysis HT12 software (FortéBio) and results were plotted in GraphPad Prism7. Experiments were repeated at least 3 times to assess consistency and control experiments with S-R, S-R/PP were performed to confirm consistency with our previous study [12].

## Cryo-EM

Grid preparation and image collection were performed for S-R/x3 and S-R/x3/D614G spike proteins essentially as described in Xiong et al. 2020 [4]. C-Flat 2/2 3C grids (Protochips) were glow-discharged for 45 seconds at 25 mA. 3 μl of freshly purified protein at 0.6 mg/ml supplemented with 0.01% octyl-glucoside (OG) was applied to the grids, which were plunge-frozen in liquid ethane using a Vitrobot (Thermo Fisher Scientific).

An aliquot of S-R/x3 freshly purified spike at 1.0 mg/ml was stored at 4°C for 40 days. 10 μl of the stored protein was subjected to plunge-freezing at 1.0 mg/ml following the same procedure as for the freshly purified S trimers. Another 10 μl of the stored protein was incubated with 1 μl of citrate acid (pH 4.8) overnight at 4°C and then plunge-frozen.

Grids were stored in liquid nitrogen and loaded into a Titan Krios electron microscope (Thermo Fisher Scientific) operated at 300 kV. Movies with 48 frames were collected with a Gatan K3 BioQuantum direct electron detector with the slit retracted. Three shots per hole were achieved with beam-image shift controlled in SerialEM-3.7.0 [34]. An accumulated dose of 50 electrons/$Å^2$ were acquired in counting mode at the magnification of 81,000 X, corresponding to a calibrated pixel size of 1.061 Å/pixel. Detailed data acquisition parameters are summarized in S1 Table.

## Cryo-EM image processing

Real-time data processing was performed in RELION-3.1's Scheduler as described in (Xiong, 2020). Motion correction, contrast transfer function (CTF) estimation, template particle picking and initial 3D classification were carried out while micrographs were being collected. An EM structure of the SARS-CoV-2 S protein in open form was filtered to 20 Å resolution as a 3D reference for template picking. Initial 3D classification with an open S model was accomplished at bin4 in batches of 500,000 particles to identify S protein trimers. Subsets of S trimers in the 3D classes which displayed clear secondary structures were pooled and subjected to one round of 2D classification cleaning. Subsequently, a second round of 3D classification was used to assess the ratio of closed and locked states (**S1 Fig**). Auto refinement, Bayesian polishing and CTF refinement were performed iteratively on classified closed and locked subsets, respectively [35,36]. Following the final round of 3D auto-refinement, map resolutions were estimated at the 0.143 criterion of the phase-randomization-corrected Fourier shell correlation (FSC) curve calculated between two independently refined half-maps multiplied by a soft-edged solvent mask. Final reconstructions were sharpened and locally filtered in RELION *post-processing* (**S2 Fig**). The estimated B-factors of each map are listed in S1 Table.

The radiation damage caused fading of the disulphide bond between the two engineered cysteines in S-R/x3 and S-R/x3/D614G. In order to recover the bond density, EM maps of closed and locked S trimers were reconstructed from the first 4 frames in the movies of S-R/x3 and the first 5 frames of S-R/x3/D614G, respectively. Distinct densities of the disulphide bond were observed in the EM structures reconstructed from early exposed frames and are shown in **Fig 2**.

## Model building and refinement

For the closed conformation structures, the SARS-CoV-2 S protein ectodomain structure (PDBID: 6ZOX [4]) was fitted into the EM density as a starting model. For the locked conformations, S structures from our previous study (PDBID: 6ZP2, 6ZOZ [4]) were used as starting models. Model building and adjustment were performed manually in Coot-0.9 [37]. Steric clash and sidechain rotamer conformations were improved using the Namdinator web server

[38]. After further manual adjustment, the locked and closed structures were refined and validated in PHENIX-1.18.261 [39] to good geometry. Refinement statistics are given in S1 Table.

## Mass spectrometry

10 μl of purified spike protein at ~ 2 mg/ml was extracted with 80 μl acetonitrile, 3 μl of organic supernatant was injected into liquid chromatography-quadrupole time of flight mass spectrometry (LC-Q-TOF-MS) for further analysis. The sample was separated using an Agilent 1290 Infinity II LC system (Agilent, Singapore) with a BEH C18 column (2.1 × 100 mm, 1.7 μm, Waters). The mobile phase consisted of (A) 0.1% formic acid in water and (B) 0.1% formic acid in acetonitrile (Optima LC/MS Grade, Fisher Chemical). The gradient was as follows: 0-1min, 5% B; 1–15 min, 5% to 95% B; 15–20 min, 95% B. The flow rate was 0.3 mL/min; column temperature, 30°C. Metabolites were detected across a mass range of m/z 100 to 1700 using a 6546 Q-TOF mass spectrometer (Agilent, Singapore). MS parameters were as follows: gas temperature, 325°C; gas flow, 8 L/min; nebulizer, 45 psig; sheath gas temperature, 325°C; sheath gas flow, 8 L/min; VCAP, 3500 V; nozzle voltage, 1500 V. AutoMS2 scan was used at 3 spectra/s, collision energy was set at 10, 20, 40 eV in collision induced dissociation (CID). Data were collected and analyzed by MassHunter 10.0 software (Agilent).

## Supporting information

**S1 Fig. Biochemical properties of purified spike proteins.** (A) Reduction of x2 disulfide bonds under native conditions by 20 and 60 min incubation with indicated concentrations of DTT, reactions were stopped by excess amount of iodoacetamide before SDS-PAGE. (B) binding of CR3022, an antibody targeting a cryptic epitope on RBD, to different spike proteins in the absence and presence of 20 mM DTT. Spike proteins were serial diluted to 200, 100, 50, 25, 12.5, 6.25, 3.13 and 0 nM and used as analytes in the assay.
(TIF)

**S2 Fig. Pipeline used for picking and classification of cryo-EM data.** Pipeline is illustrated for S-R/D614G, S-R/x3, S-R/x3/D614G, S-R/x3 after 40 days, and the latter after transfer to pH5.0 buffer. After automated picking, 3D and 2D classification steps were used to remove contaminating objects. 3D classification was then used to sort the data into locked and closed conformations.
(TIF)

**S3 Fig. Cryo-EM densities of x3 disulfide bonds.** (A)—(D), left panels show the locations of the x3 disulfide bonds within the corresponding spike trimer structures; right panels show molecular models and densities of the x3 disulfide bonds. The radiation damage caused fading of the disulphide bond between the two engineered cysteines in the S-R/x3 and S-R/x3/D614G final maps. In order to recover the disulfide bond densities, EM maps of closed and locked S trimers were reconstructed from the first 4 frames in the movies of S-R/x3 and the first 5 frames of S-R/x3/D614G, respectively. Distinct densities of the disulfide bond were observed in the EM structures reconstructed from early exposed frames and are shown the right panels.
(TIF)

**S4 Fig. Resolution assessment of cryo-EM structures.** (A) Local resolution maps for all structures calculated using RELION. (B) Global resolution assessment by Fourier shell correlation at the 0.143 criterion (left panel), and correlations of model vs map by Fourier shell correlation at the 0.5 criterion (right panel).
(TIF)

**S5 Fig. Structural features of S-R/x3 and S-R/x3/D614G S proteins.** (A)—(F), top views of S-R/x3 and S-R/x3/D614G spikes in locked-1, locked-2 and closed conformational states. The three-fold axes are indicated by triangles. The structural elements that are closest to the three-fold axes in the locked structures are indicated by dots (panels (A), (B), (D), (E)). The same structural elements are indicated by dots in the closed conformations to highlight the movement of the receptor binding domains between the two conformational states (panels (C) and (F)). (G) comparison of S-R, S-R/x3 and S-R/x3/D614G spikes in closed conformations. Left panel, overlay of the three structures, the box indicates the zoomed-in view in the right panel. Right panel, a movement of 3.7 Å was observed between the closed S-R (without x3 disulphide) and the closed S-R/x3 and S-R/x3/D614G spikes (with x3 disulphide).
(TIF)

**S6 Fig. Identification of biliverdin IXα by mass spectrometry.** (A) extracted-ion chromatogram (EIC) of organic extract of purified spike protein identifies a prominent peak with an elution time of 9.185 min. (B) MS analysis of the peak in panel a identifies a major ion species with an m/z of 583.2551 consistent to the structure of biliverdin IXα (panel (D)) with a mass difference of -0.53 ppm. (C) MS/MS spectrum of biliverdin IXα in the sample identifies the parent ion (indicated by the blue diamond) and a fragment with an m/z of 297.1228 corresponding to the proposed fragment ion of biliverdin IXα (indicated by the dashed line in panel (D)). (D) Chemical structure of biliverdin Ixα and proposed fragmentation pattern in MS/MS (dashed line).
(TIF)

**S7 Fig. Comparisons of spike proteins in different conformations.** (A)—(C), S1 parts of various different spike structures were aligned using domain D to highlight the movement of the RBD and domain C between the different structures.
(TIF)

**S8 Fig. Domain D and surrounding structural features of S-R/x3 locked-1, S-R/x3/D614G locked-2, S-R/x3/D614G closed, S-R/x3 low-pH locked-2 structures compared to different deposited structures.** (A)—(H), S-R/x3 locked-1 (colored) was compared to various locked SARS-CoV-2 spike structures (gray), in most of these structures, the domain D and surrounding area adopt a similar structure as in the S-R/x3 locked-1 structure. (I)—(J), The full-length D614G SARS-CoV-2 spike structure (gray, PDB: 7krq) was compared to S-R/x3/D614G locked-2 and S-R/x3/D614G closed structures (colored) to show the structure is in a intermediate state between locked and closed conformations. (K) S-R/x3/D614G closed structure compared to the full-length D614G SARS-CoV-2 spike in open conformation. (L) S-R/x3 low-pH locked-2 structure compared to another deposited low-pH SARS-CoV-2 spike structure (6xm5, chain A), which shows the most well-defined densities for regions around domain D among their low-pH structures (6xm4, 6xm5, 6xlu, 7jwy) [23]. PDB-IDs are from the following studies: 7jjj, 7iii [8]; 6zb4, 6zb5 [10]; 6zge, 6zgi [21]; 7ddd [14]; 6xr8 [9]; 7krq, 7qrr [22]; 6xm5 [23].
(TIF)

**S1 Table. Cryo-EM data collection, refinement and validation statistics.**
(PDF)

# Acknowledgments

We thank the staff of the MRC-LMB EM Facility their support; Jake Grimmett and Toby Darling for scientific computing infrastructure; and Patricia Edwards for supporting cell culture.

## Author Contributions

**Conceptualization:** Kun Qu, Xiaoli Xiong, John A. G. Briggs.

**Funding acquisition:** Andrew P. Carter, Xiaoli Xiong, John A. G. Briggs.

**Investigation:** Kun Qu, Qiuluan Chen, Katarzyna A. Ciazynska, Banghui Liu, Xixi Zhang, Jingjing Wang, Yujie He, Jiali Guan, Jun He, Tian Liu, Xiaofei Zhang, Xiaoli Xiong.

**Methodology:** Kun Qu, Qiuluan Chen, Xiaoli Xiong, John A. G. Briggs.

**Project administration:** Xiaoli Xiong, John A. G. Briggs.

**Resources:** Andrew P. Carter, Xiaoli Xiong, John A. G. Briggs.

**Supervision:** Xiaoli Xiong, John A. G. Briggs.

**Visualization:** Kun Qu, Qiuluan Chen, Xiaoli Xiong, John A. G. Briggs.

**Writing – original draft:** Kun Qu, Xiaoli Xiong, John A. G. Briggs.

**Writing – review & editing:** Qiuluan Chen, Andrew P. Carter.

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
