## [Decision Letter · Decision Letter 0]

18 Mar 2022

Dear Briggs,

Thank you very much for submitting your manuscript "Engineered disulfide reveals structural dynamics of locked SARS-CoV-2 spike" for consideration at PLOS Pathogens. As with all papers reviewed by the journal, your manuscript was reviewed by members of the editorial board and by several independent reviewers. The reviewers appreciated the attention to an important topic. We are sorry this process took so long. Based on the reviews, we are likely to accept this manuscript for publication, providing that you modify the manuscript according to the review recommendations.

Sincerely,

Daved H Fremont

Associate Editor

PLOS Pathogens

Andrew Pekosz

Section Editor

PLOS Pathogens

Kasturi Haldar

Editor-in-Chief

PLOS Pathogens

orcid.org/0000-0001-5065-158X

Michael Malim

Editor-in-Chief

PLOS Pathogens

orcid.org/0000-0002-7699-2064

Reviewer Comments (if any, and for reference):

Reviewer's Responses to Questions

**Part I - Summary**

Reviewer #1: The manuscript from Qu et al investigates the locked conformation of the SARS-CoV-2 spike, a conformation that is infrequently observed by cryo-EM studies of spike protein as compared to the closed or open conformations. The locked conformation has not been observed in high-resolution structures of spikes on virions, however, raising some concern as to whether the locked conformation is an in vitro artifact. To more easily study the locked conformation, the authors design a disulfide bond by substitution D427 and V987 with cysteine. Biochemical characterization, including non-reducing SDS-PAGE, confirms that the disulfide bond does form as intended and prevents ACE2 binding, as expected. Cryo-EM analysis of this variant shows that 76% of the particles are in the locked conformation and 23% are in the closed conformation. Data processing is able to discriminate two different locked conformations, named locked-1 and locked-2, that differ primarily by the position of residues in domain D. Introduction of the D614G substitution shifts the distribution of spikes toward the closed conformation, suggesting that D614G makes the spikes inherently less stable and more prone to open, which is consistent with data and conclusions published in 2020.

The manuscript has several strengths. The design of a disulfide-stabilized locked spike, or mostly locked spike, is interesting, and may be of use to the field. The biochemical and structural characterizations are very thorough and are performed to a high standard. The writing, although technical in nature, is clear. The major weakness or limitation of the manuscript is the potential lack of physiological relevance. As noted by the authors and mentioned above, the locked state has not been observed in structures of spikes on the surface of virions. A potential explanation is that the locked state is some artifact of recombinantly expressed and purified spike protein. The authors primarily speculate about the functional role of the locked-1 and locked-2 states (lines 359 and 387), but do not experimentally support their claims. And even if the locked state is sampled by spikes on infectious virions in vivo, it is possible that the different locked-1 and locked-2 states observed in these studies is due to the artificial nature of the introduced disulfide bond. Overall, it is solid work, but may be too preliminary at this time.

Reviewer #2: Investigators have advanced prior studies of SARS-CoV-2 spike structures by “locking” spike ectodomains with an engineered (aa 427 to aa 987) disulfide bridge. Biochemical tests revealed the disulfide locked ectodomains did not display RBDs to ACE2 (RBD-down) but RBDs could be raised for ACE2 binding by DTT reduction. Cryo-EM showed that most disulfide locked trimers were in closed conformations. Further analysis revealed two distinct locked conformations. The D614G change reduced the proportion of locked conformations. These are important results that advance understanding of spike protein structures in different conformations.

Prolonged incubation of the locked trimers resulted in conversion to a “closed” conformation, and then exposure of the closed form to pH5 re-locked the trimers. These changes regulating locked to closed transitions involved specific subdomains, notably the NTD associated domains. The manuscript includes impressive detail on the structural differences between the locked and closed conformations, and the effects of a D614G substitution on these differences.

Structures are used to infer models of spike dynamics during virus secretion. The suggestion is that virus spikes are assembled into locked conformations during virus assembly and egress. This is based on the assumption that viruses egress through acidic organelles, which lock the trimers. Once secreted, virus spikes at neutral pH slowly convert to closed conformations, which are prerequisite to RBD opening.

The models are very intriguing, however, it is recommended that they be further considered. The results of Ghosh et al. indicate that organelles through which the viruses egress are at neutral pH – there are accessory proteins of the virus that neutralize endolysosome lumenal compartments. Furthermore, the natural spikes undergo furin cleavage in the TGN, which effects conformational changes that were not described in the models. In addition, the D614G change that is highlighted in the model is known to provide a stabilizing force, maintaining the S1-S2 heterodimers. It is difficult to appreciate this important S1-S2 stabilizing property in the context of the model involving locked and closed conformations. Finally, the notion that there are very slow transitions from the locked to closed, and then to open conformations of spike do not appear consistent with the very rapid cell membrane fusions that take place when spike-expressing cells are incubated with ACE2-expressing target cells. In this case, as soon as spikes appear on plasma membranes, they promote cell membrane fusions, suggesting that they are in open conformations.

**Part II – Major Issues: Key Experiments Required for Acceptance**

Reviewer #1: (No Response)

Reviewer #2: The models are very intriguing, however, it is recommended that they be further considered. The results of Ghosh et al. indicate that organelles through which the viruses egress are at neutral pH – there are accessory proteins of the virus that neutralize endolysosome lumenal compartments. Furthermore, the natural spikes undergo furin cleavage in the TGN, which effects conformational changes that were not described in the models. In addition, the D614G change that is highlighted in the model is known to provide a stabilizing force, maintaining the S1-S2 heterodimers. It is difficult to appreciate this important S1-S2 stabilizing property in the context of the model involving locked and closed conformations. Finally, the notion that there are very slow transitions from the locked to closed, and then to open conformations of spike do not appear consistent with the very rapid cell membrane fusions that take place when spike-expressing cells are incubated with ACE2-expressing target cells. In this case, as soon as spikes appear on plasma membranes, they promote cell membrane fusions, suggesting that they are in open conformations.

**Part III – Minor Issues: Editorial and Data Presentation Modifications**

Reviewer #1: (No Response)

Reviewer #2: (No Response)

PLOS authors have the option to publish the peer review history of their article (what does this mean?). If published, this will include your full peer review and any attached files.

Reviewer #1: No

Reviewer #2: No

Figure Files:

Data Requirements:

Reproducibility:

References:

---

## [Editor Report · Decision Letter 1]

9 May 2022

Dear Briggs,

We are pleased to inform you that your manuscript 'Engineered disulfide reveals structural dynamics of locked SARS-CoV-2 spike' has been provisionally accepted for publication in PLOS Pathogens.

Best regards,

Daved H Fremont

Associate Editor

PLOS Pathogens

Andrew Pekosz

Section Editor

PLOS Pathogens

Kasturi Haldar

Editor-in-Chief

PLOS Pathogens

orcid.org/0000-0001-5065-158X

Michael Malim

Editor-in-Chief

PLOS Pathogens

orcid.org/0000-0002-7699-2064
---

## [Editor Report · Acceptance letter]

26 Jul 2022

Dear Briggs,

We are delighted to inform you that your manuscript, "Engineered disulfide reveals structural dynamics of locked SARS-CoV-2 spike," has been formally accepted for publication in PLOS Pathogens.

Best regards,

Kasturi Haldar

Editor-in-Chief

PLOS Pathogens

orcid.org/0000-0001-5065-158X

Michael Malim

Editor-in-Chief

PLOS Pathogens

orcid.org/0000-0002-7699-2064